

# Quantitative evaluation of muscle mass based on chest high-resolution CT and its prognostic value for tuberculosis: a retrospective study

Ankang Huang[1,2], Yuyao Zhang[1,2], Qi Dai[1], Jingfeng Zhang[1] and Jianjun Zheng[1]

[1] Department of Radiology, Ningbo No.2 Hospital, Ningbo, Zhejiang, China
[2] School of Medicine, Shaoxing University, Shaoxing, Zhejiang, China

Corresponding author
Jianjun Zheng, zhjjnb2@163.com

## ABSTRACT

**Objective**. This study aims to explore the prognostic value of quantitatively evaluating muscle mass using chest high resolution computed tomography (HRCT) in patients with active tuberculosis (TB).

**Methods**. This retrospective cohort study collected data from 309 patients with active TB diagnosed at Ningbo No.2 Hospital from 2020 to 2023. Based on the skeletal muscle index (SMI) at the T12 vertebra (with thresholds of $<28.8$ cm$^2$/m$^2$ for men and $<20.8$ cm$^2$/m$^2$ for women), patients were divided into a low muscle mass group and a normal muscle mass group. The study compared baseline characteristics, muscle mass-related indicators, body mass index (BMI), and imaging features between the two groups. The correlation between muscle mass-related indicators, BMI, and TB imaging features and prognosis was analyzed. Receiver operating characteristic (ROC) curve analysis and multivariate logistic regression were used to assess the prognostic value of muscle mass-related indicators and BMI in patients undergoing anti-TB treatment.

**Results**. A total of 309 patients were included in the study, divided into a normal muscle mass group ($n = 229$) and a low muscle mass group ($n = 80$). There was a significant difference in prognosis between the two groups ($\chi^2$ test, $p < 0.05$). Patients in the low muscle mass group were older, had a higher proportion of males, and had a lower BMI ($p < 0.05$). Additionally, these patients had a higher likelihood of developing pulmonary cavities ($p < 0.05$). In terms of imaging features, the two groups showed significant differences in the pre-treatment proportion of pulmonary fibrotic bands, ground-glass opacities, consolidation, lesion percentage, and lesion absorption ratio (all $p < 0.05$). Univariate analysis indicated that both the T12 skeletal muscle index (T12 SMI) and BMI were correlated with TB imaging characteristics ($p < 0.05$), with T12 SMI showing a stronger correlation than BMI. Multivariable linear regression analysis revealed that after adjusting for age, gender, and T12 skeletal muscle radiation attenuation (T12 SMRA), T12 SMI remained significantly correlated with the whole-lung lesion proportion ($\beta : -4.56$, 95% CI [$-5.45$ to $-3.67$]) and lesion absorption ratio ($\beta : 0.036$, 95% CI [$0.031$–$0.041$]). Multivariable logistic regression analysis demonstrated that after accounting for age, gender, T12 SMRA, T12 SMI was significantly associated with the prognosis of TB patients (OR: 20.10, 95% CI [$8.81$–$51.56$], $p < 0.05$), indicating that low T12 SMI is an independent risk factor associated with poor prognosis. ROC curve analysis indicated that T12 SMI may offer advantages over BMI, with an area

under the ROC curve (AUC) of T12 SMI (0.761, 95% CI [0.690–0.832]) higher than the AUC of BMI (0.700, 95% CI [0.619–0.781].

**Conclusion**. Quantitative evaluation of muscle mass using chest HRCT, particularly the T12 SMI, may provide valuable prognostic information for tuberculosis patients, potentially offering advantages over BMI in assessing patient outcomes.

# INTRODUCTION

*Mycobacterium tuberculosis*, the bacterium responsible for tuberculosis (TB), is regarded as one of the most lethal infectious agents worldwide (*Sinha et al., 2019*). China remains one of the countries with the highest TB burden (*Reid et al., 2023*). According to the WHO Global Tuberculosis Report 2023, the estimated number of new TB cases in China in 2022 is 748,000, accounting for 7.1% of the global total and placing the country third in TB incidence worldwide (*World Health Organization, 2023*). Malnutrition is a significant population-level risk factor for TB (*Sinha et al., 2021*), with numerous studies consistently demonstrating its association with increased incidence of pulmonary TB, greater disease severity, poorer treatment outcomes, and higher mortality rates (*Lonnroth et al., 2010*; *Niki et al., 2020*; *Ma et al., 2022*; *Jeong et al., 2023*). Therefore, accurate assessment of patients' nutritional status and TB severity, along with early prediction of treatment outcomes, is essential for effective clinical management, the development of nutritional interventions, and guidance on lifestyle modifications.

The 2018 Global Leadership Initiative on Malnutrition (GLIM) consensus has identified low body mass index (BMI) and reduced muscle mass as key phenotypic criteria for diagnosing malnutrition (*Cederholm et al., 2019*). Previous studies have consistently shown a correlation between low BMI and worse outcomes in TB patients, with those having a lower BMI often presenting more severe disease and poorer prognosis (*Hoyt et al., 2019*; *Min et al., 2023*; *Sinha et al., 2023*). However, BMI has notable limitations as it does not differentiate between fat, muscle, water, or bone content. Research indicates that malnutrition reduces skeletal muscle mass (*Barazzoni et al., 2022*), and measuring muscle mass provides a more comprehensive evaluation of body composition, potentially leading to more accurate assessments of patients' nutritional status. Sarcopenia, or muscle loss, has been identified as a risk factor for mortality in elderly TB patients and can predict TB-related mortality (*Kakiuchi et al., 2024*). Elderly individuals with sarcopenia are also at higher risk for developing TB (*Yoo et al., 2021*), while the risk of sarcopenia increases in TB patients with incomplete pulmonary lesion absorption (*Choi et al., 2017*). These findings highlight the significant role of muscle mass in both TB development and prognosis. Although the causal relationship remains unclear (*Karakousis, Gourgoulianis & Kotsiou, 2023*), evidence suggests that nutritional interventions to address low muscle mass could improve TB patient outcomes (*Shin et al., 2021*). While existing studies have explored the muscle mass-TB relationship, they typically rely on additional testing or questionnaires,

which may place a financial and time burden on patients. Furthermore, most research has focused solely on elderly TB patients, limiting generalizability. To date, there is a lack of comparative studies on the predictive value of BMI and muscle mass indicators for TB prognosis.

To address these research gaps, this cohort study conducted between 2020 and 2023 quantitatively measures muscle mass and pulmonary tuberculosis computed tomography (CT) imaging features using high-resolution CT. We analyze the correlation between muscle mass and chest CT characteristics in tuberculosis patients and investigate the predictive value of muscle mass for anti-tuberculosis treatment outcomes, aiming to inform clinical interventions for TB patients by identifying those at higher risk of poor prognosis based on muscle mass evaluation. Given that routine chest CT scans typically extend only to the T12 vertebral level and do not capture the L3 level, T12 SMI serves as a practical and reliable alternative for assessing muscle mass in tuberculosis patients, avoiding the need for additional imaging and minimizing radiation exposure.

## METHODS

### Study population

A total of 3,905 TB patients treated in Ningbo No.2 Hospital hospital between January 2020 and August 2023 were screened based on specific inclusion and exclusion criteria. The inclusion criteria were: (1) TB patients meeting the bacteriological confirmation criteria according to the WHO consolidated guidelines on tuberculosis (2024 edition) (*World Health Organization, 2024*), who were newly treated and drug-sensitive; (2) patients who received the standard quadruple anti-tuberculosis therapy (2HRZE/4HR); (3) patients who underwent CT scans before treatment and six months post-treatment; and (4) clear chest CT images with thin-section scans (<2 mm). The exclusion criteria were: (1) patients with conditions that significantly affect chest CT results (*e.g.*, lung cancer, defibrillator or pacemaker placement, history of cardiac or pulmonary surgery); (2) incomplete clinical records; (3) AI-assisted quantification errors in lung lesion analysis; (4) significant pleural effusion; (5) TB patients with comorbidities such as diabetes, HIV, or COPD; and (6) irregular anti-tuberculosis medication or treatment interruption resulting in incomplete therapy (Fig. 1). After applying these criteria, 309 patients were included in the final cohort, comprising 216 males and 93 females, aged 18 to 93 years, with a median age of 43. Patients were further categorized into two groups using the T12 Skeletal Muscle Index (T12 SMI) as a standard (T12 SMI <28.8 $cm^2/m^2$ for males and <20.8 $cm^2/m^2$ for females) (*Derstine et al., 2018*): the low muscle mass group (80 patients) and the normal muscle mass group (229 patients). This study intentionally focused on drug-sensitive tuberculosis patients without major comorbidities (*e.g.*, diabetes, HIV, COPD) to isolate the impact of muscle mass on TB prognosis in a relatively homogeneous population, thereby providing clearer insights into the relationship between muscle mass and TB outcomes in the absence of confounding factors from other chronic conditions. The Medical Ethics Committee of Ningbo No.2 Hospital in Zhejiang Province granted ethical clearance for this investigation (Ethics Approval No.:NBEY-2023-064). This study was retrospective and received a waiver of patient consent.

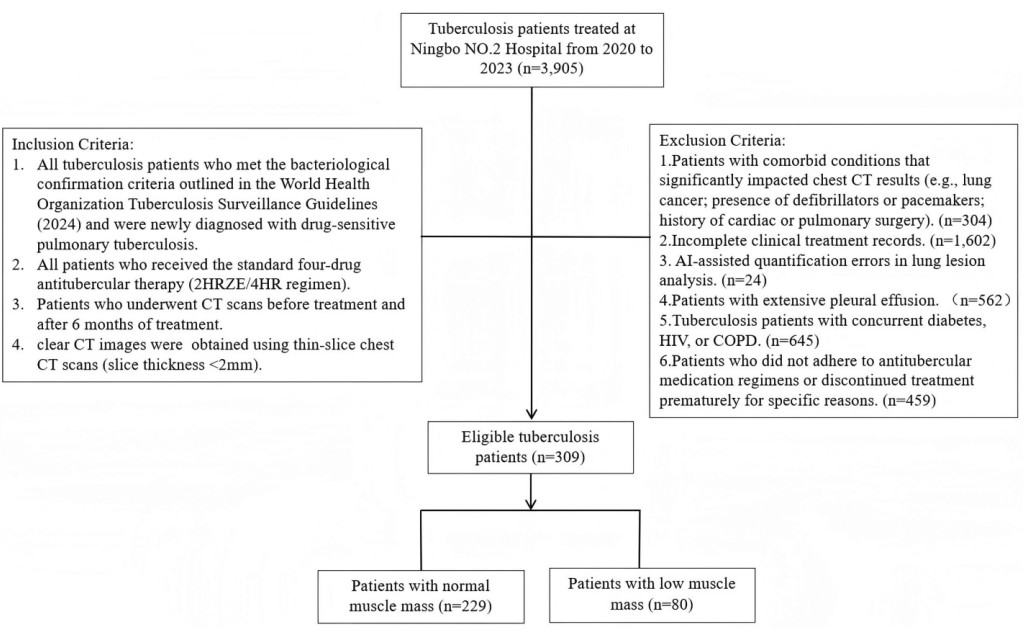

**Figure 1** Flowchart of study population.

## Quantitative assessment
### Quantitative assessment of pulmonary imaging features

Automated AI-assisted quantitative analysis of full-lung imaging features was conducted using AVIEW software. Typical TB signs such as consolidation, fibrosis, ground-glass opacity, and cavitation were automatically extracted and quantified as a proportion of the total lung volume. The whole lung lesion proportion before treatment was defined as the percent of the total lung volume occupied by the lesion. The whole lung lesion absorption ratio was calculated as: (Total Lung Lesion Volume Before Treatment–Total Lung Lesion Volume After Treatment)/(Total Lung Lesion Volume Before Treatment). Quantitative analysis was performed on chest CT scans both before treatment and at six months post-treatment (Fig. 2).

To validate the AI-assisted quantification of lung lesions, all chest CT images analyzed using the AVIEW software were independently reviewed by two radiologists specialized in thoracic imaging, each with 15 and 12 years of experience, respectively. Both radiologists were blinded to each other's assessments as well as to the clinical data of the patients when verifying the AI outputs. The review process included an assessment of the software's accuracy in quantifying lesion extent and classification. If any discrepancies or errors were found in the AI-generated results, the case was excluded from the study. Only cases where both radiologists agreed that the quantification results were accurate were included in the final analysis. Throughout the review process, both radiologists were blinded to the patients' clinical data and diagnostic information. In total, 24 patients were excluded due to AI quantification errors. The chest CT images of the final cohort of 309 patients were independently re-verified by the radiologists, confirming that the AI-assisted
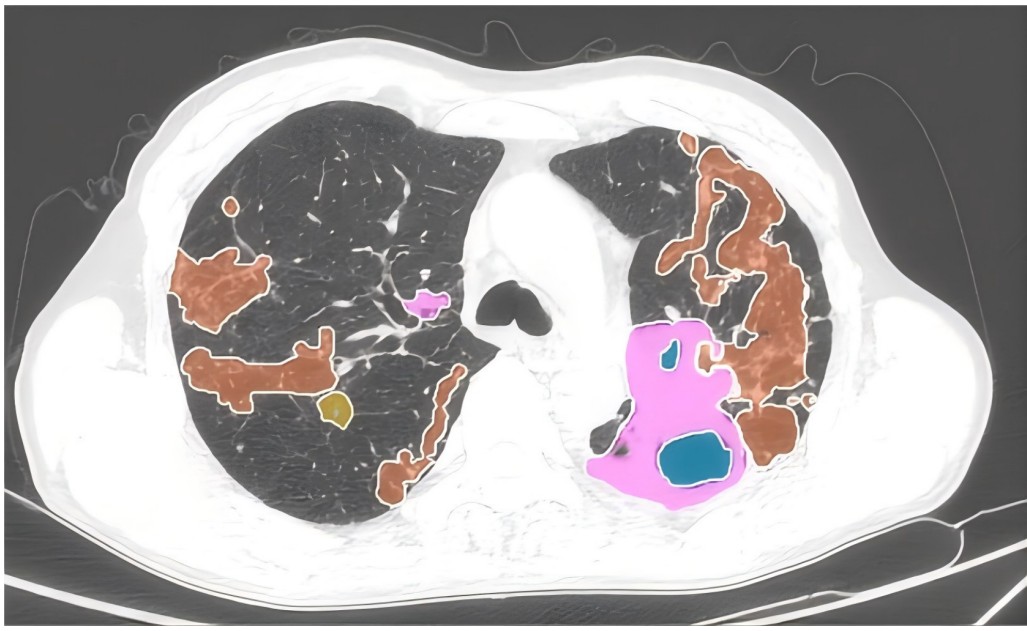

**Figure 2** **Quantitative assessment of pulmonary imaging characteristics.** The CT imaging features of the lungs, highlighting various pathological changes using different colors for easy identification. Pink represents areas of consolidation. Orange depicts fibrous strands, showing the presence of linear or band-like structures within the lung tissue. Blue indicates the presence of cavities, which are hollow spaces within the lung. Yellow highlights ground-glass opacities, which are hazy areas on the CT scan.

quantification results were accurate. To assess the reproducibility of the software's results, a repeat quantification analysis was performed on the pre-treatment chest CT images of the 309 patients. Intraclass correlation (ICC) analysis was conducted on the proportion of total lung lesions, yielding a high ICC value of 0.947 95% CI [0.934–0.957].

### Quantitative assessment of muscle mass

T12 vertebral level images from chest CT scans were manually delineated and analyzed using sliceOmatic software to calculate the cross-sectional area and mean density of all skeletal muscles. Two consecutive slices were evaluated, and the average values were determined. Muscle tissue was identified using standardized Hounsfield unit (HU) thresholds (−29 to +150 HU) (*Albano et al., 2020*). The T12 Skeletal Muscle Index (T12 SMI) was calculated by dividing the total cross-sectional area of skeletal muscle at the T12 vertebral level (cm$^2$) by the square of the patient's height (m$^2$). Muscle mass was assessed using both the T12 SMI and T12 skeletal muscle radiation attenuation (T12 SMRA). For each patient, the T12 SMI and T12 SMRA were obtained from pre-treatment chest CT scans performed before anti-tuberculosis therapy. See Fig. 3 for reference.

### HRCT examination

All patients underwent high-resolution chest CT scans using the Apsaras CT machine from Konida Corporation. Patients were positioned supine, head first, with the scan range from the thoracic inlet to the diaphragm. Scanning parameters were one mm slice thickness,

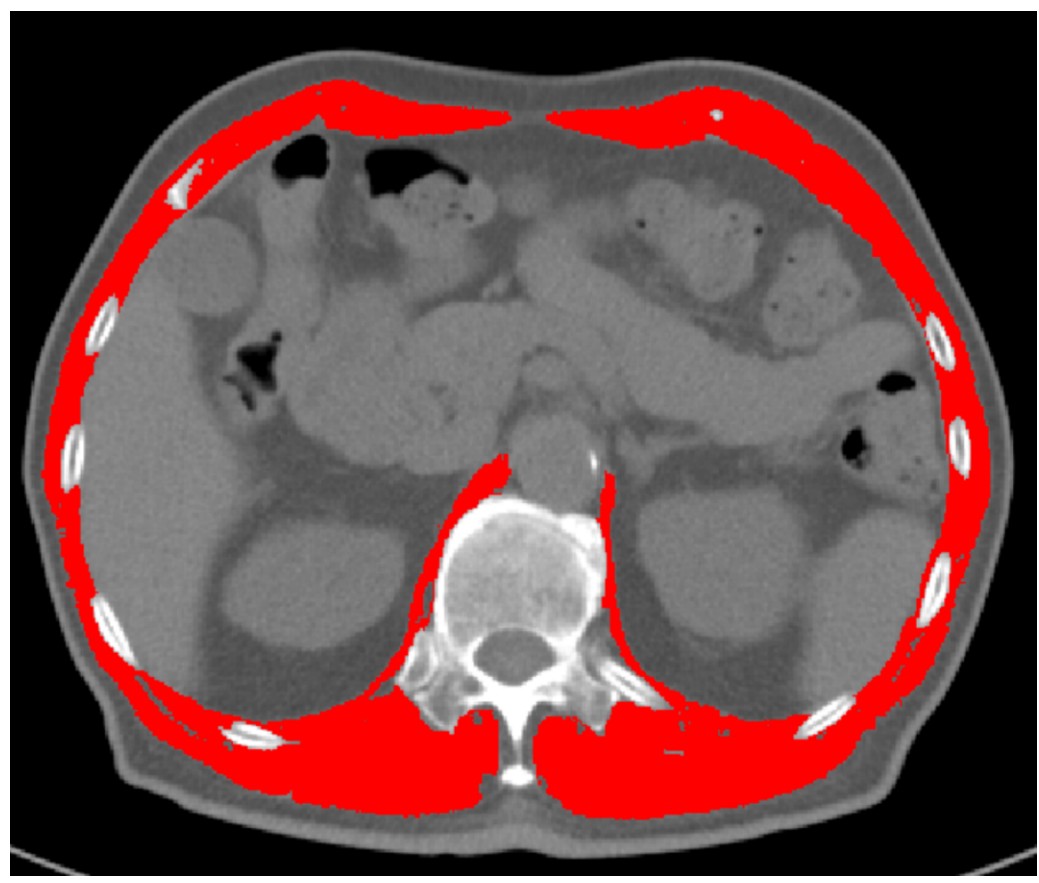

**Figure 3 Delineation and quantification of skeletal muscle at the T12 vertebral level using sliceOmatic software.** This figure illustrates the delineation of skeletal muscle at the T12 vertebral level using the sliceOmatic software. The software automatically calculates the area and assesses the density of the skeletal muscle within the region of interest (ROI). The skeletal muscle highlighted in red represents the areas identified and marked by the software.

120 kV tube voltage, 110 mAs tube current, one mm pitch, 512 × 512 matrix, and 0.75 s rotation speed.

## Prognosis evaluation

The outcomes were evaluated based on the standards outlined in the World Health Organization Consolidated Guidance on Tuberculosis Data Generation and Use (2024) (*World Health Organization, 2024*). Outcome was classified into five categories: cure, treatment completion, treatment success, treatment failure, and death. Among these, cure, treatment completion, and treatment success were grouped as favorable prognosis, while treatment failure and death were categorized as poor prognosis. To be specific, cure was defined as pulmonary tuberculosis patients who were bacteriologically confirmed at the start of treatment, completed therapy according to national policy recommendations, exhibited evidence of bacteriological response, and showed no signs of treatment failure. Treatment completion referred to patients who completed therapy as per national guidelines but did

not meet the criteria for cure or failure. Treatment success encompassed those who were either cured or completed treatment as defined above. Treatment failure was characterized by the need to terminate or permanently alter the treatment regimen. Death referred to patients who died before or during treatment for any reason.

## Statistical analysis

Data were analyzed using R version 4.3.3. For continuous variables that were normally or approximately normally distributed, mean ± standard deviation was used for statistical description, and independent sample t-tests were used for between-group comparisons; Pearson's correlation was applied for correlation analysis. For non-normally distributed data, the median (p25, p75) was used for statistical description, independent sample rank-sum tests for group comparisons, and Spearman's method for correlation analysis. Multivariable linear regression was used to study the association between the skeletal muscle index and imaging indicators related to whole lung lesion (whole lung lesion proportions before treatment, whole lung lesion absorption ratio). Multivariable logistic regression was used to investigate whether the skeletal muscle index was an influencing factor for poor prognosis. To evaluate multicollinearity, we calculated the variance inflation factor (VIF), with the results presented in Tables S1–S3. To explore whether the relationship between T12 SMI and prognosis varies by key demographic factors, we conducted subgroup analyses by gender, age group ($\leq$45, >45 years old), smoking and drinking. The effect of the skeletal muscle index in predicting prognosis was evaluated using ROC curve analysis, calculating the area under the curve (AUC), sensitivity, specificity, and other metrics. The cut-off values in these analyses were determined by maximizing the Youden index, a standard method for identifying optimal thresholds in diagnostic tests. Calculated as (Sensitivity + Specificity $-1$), the Youden index identifies the cut-off value that achieves the best balance between sensitivity and specificity, ensuring thresholds that optimize overall diagnostic performance. Statistical significance of the difference between AUC using various indicators was calculated by Delong's test (Table S4). All tests were two-sided, with $p < 0.05$ considered statistically significant.

# RESULTS

## Baseline characteristics of patients

Comparison between the normal muscle mass group ($n = 229$) and the low muscle mass group ($n = 80$) revealed significant differences in prognosis according to the chi-square test ($p < 0.05$). The low muscle mass group had a higher average age, a lower proportion of females, and a lower BMI, all of which were statistically significant ($p < 0.05$). There were no significant differences between the two groups in terms of smoking, alcohol consumption, or educational level. The low muscle mass group exhibited a higher incidence of cavitation, a greater number of affected lung lobes before treatment, and higher proportions of pre-treatment whole-lung fibrotic lesions, ground-glass opacities, and consolidations compared to the normal muscle mass group, all of which were statistically significant ($p < 0.05$). The normal muscle mass group demonstrated a higher proportion of lesion

absorption (difference between pre-treatment whole-lung lesion proportion and post-treatment whole-lung lesion proportion/pre-treatment whole-lung lesion proportion) than the low muscle mass group, with statistical significance ($p < 0.05$) (Table 1).

## Correlation analysis of T12 SMI, T12 SMRA, and BMI with radiological features of tuberculosis

### Univariate correlation analysis

Spearman correlation analysis was used to examine the relationships between T12 SMI, T12 SMRA, BMI, and radiological features of tuberculosis. The T12 SMI was found to be negatively correlated with the pre-treatment proportion of whole-lung fibrotic lesions, ground-glass opacities, consolidations, and the overall lung lesion proportion, as well as positively correlated with the proportion of lesion absorption. Specifically, the skeletal muscle index was strongly negatively correlated with the pre-treatment whole-lung lesion proportion ($r = -0.870$, $p < 0.001$), and moderately to strongly negatively correlated with the pre-treatment proportions of whole-lung consolidations and fibrotic lesions ($r = -0.606, -0.701$, $p < 0.001$), and moderately negatively correlated with the proportion of ground-glass opacities ($r = -0.512$, $p < 0.001$). A strong positive correlation was observed between T12 SMI and the proportion of lesion absorption ($r = 0.717$, $p < 0.001$). BMI was also negatively correlated with the pre-treatment whole-lung lesion proportion, pre-treatment proportions of consolidations, fibrotic lesions, and ground-glass opacities ($r = -0.571, -0.363, -0.474, -0.361$, $p < 0.001$), and positively correlated with the proportion of lesion absorption ($r = 0.486$, $p < 0.001$). T12 SMRA showed little or no correlation with the radiological features of tuberculosis. These results indicate that the correlation between T12 SMI and radiological features of tuberculosis was stronger than that of BMI, while T12 SMRA showed no significant correlation with the radiological features of tuberculosis (Fig. 4).

### Multivariable association analysis

To further clarify the relationship between muscle mass and the severity and outcomes of tuberculosis, T12 SMI, which showed a strong correlation in univariate analysis, was included in a multiple linear regression analysis. After controlling for age, gender, and T12 vertebral skeletal muscle density, the skeletal muscle index remained negatively correlated with the pre-treatment whole-lung lesion proportion ($\beta$: $-4.56$, 95% CI [$-5.45$ to $-3.67$], $p < 0.001$) and positively correlated with the proportion of lesion absorption ($\beta$: 0.036, 95% CI [0.031–0.041], $p < 0.001$) (Tables 2 and 3).

## Factors influencing poor prognosis in tuberculosis patients

Among 309 patients, 47 experienced poor outcomes. Multivariable logistic regression analysis showed that after controlling for age, gender, and T12 vertebral skeletal muscle density, T12 skeletal muscle mass index was associated with prognosis (OR = 20.095, 95% CI [8.808–51.554], $p < 0.001$). Low T12 SMI is associated with poor prognosis in tuberculosis patients. Additionally, age (OR = 1.028, 95% CI [1.002–1.055], $p < 0.035$) and female gender (OR = 3.084, 95% CI [1.116–8.968], $p < 0.032$) were also identified as independent risk factors associated with the prognosis in tuberculosis patients (Table 4).

**Table 1  Comparison of baseline characteristics between patients with normal muscle mass and low muscle mass.**

| Variable | Normal muscle mass group (*n* = 229) | Low muscle mass group (*n* = 80) | t/Z/$\chi^{2a}$ | p |
|---|---|---|---|---|
| Prognosis, n (%) | | | 68.175 | <0.001 |
| Good | 217 (94.8) | 45 (56.2) | | |
| Poor | 12 (5.2) | 35 (43.8) | | |
| Age, Median (Q1, Q3) | 41.00 (29.00, 58.00) | 52.00 (32.00, 64.25) | −2.107 | 0.035 |
| Gender, n (%) | | | 13.711 | <0.001 |
| Male | 147 (64.2) | 69 (86.2) | | |
| Female | 82 (35.8) | 11 (13.8) | | |
| Weight, Median (Q1, Q3) | 60.00 (52.00, 65.00) | 50.50 (45.75, 57.00) | 5.929 | <0.001 |
| Height, Median (Q1, Q3) | 1.67 (1.60, 1.72) | 1.70 (1.65, 1.74) | −2.609 | 0.009 |
| BMI, Mean ± SD | 21.42 ± 2.69 | 17.87 ± 1.97 | 12.529 | <0.001 |
| Alcohol Consumption, n (%) | | | 0.007 | 0.935 |
| No | 188 (82.1) | 66 (82.5) | | |
| Yes | 41 (17.9) | 14 (17.5) | | |
| Smoking, n (%) | | | 1.367 | 0.242 |
| No | 178 (77.7) | 57 (71.2) | | |
| Yes | 51 (22.3) | 23 (28.7) | | |
| Education Level, n (%) | | | 0.062 | 0.803 |
| Below High School | 128 (55.9) | 46 (57.5) | | |
| High School or Above | 101 (44.1) | 34 (42.5) | | |
| T12 Vertebral Skeletal Muscle Area, Mean ± SD | 87.15 ± 19.86 | 72.05 ± 12.52 | 7.874 | <0.001 |
| T12 Skeletal Muscle Radiation Attenuation (SMRA), Median (Q1, Q3) | 43.96 (38.52, 48.19) | 44.48 (38.25, 48.09) | −0.325 | 0.745 |
| T12 Vertebral Skeletal Muscle Index (SMI), Mean ± SD | 31.19 ± 5.21 | 24.98 ± 3.14 | 12.614 | <0.001 |
| Presence of Cavities, n (%) | | | 15.078 | <0.001 |
| No | 174 (76.3) | 42 (53.2) | | |
| Yes | 54 (23.7) | 37 (46.8) | | |
| Number of Cavities Before Treatment, Median (Q1, Q3) | 2.00 (1.00, 3.00) | 3.00 (1.00, 7.00) | −1.089 | 0.276 |
| Number of Affected Lung Lobes Before Treatment, Median (Q1, Q3) | 1.00 (1.00, 1.00) | 1.00 (1.00, 2.00) | −2.155 | 0.031 |
| Whole Lung Fibrous Strands Before Treatment, Median (Q1, Q3) (%) | 0.28 (0.09, 1.30) | 1.47 (0.68, 3.16) | −6.784 | <0.001 |
| Whole Lung Ground-glass Opacity Before Treatment, Median (Q1, Q3) (%) | 0.21 (0.05, 1.15) | 0.80 (0.21, 3.97) | −4.345 | <0.001 |
| Whole Lung Consolidation Before Treatment, Median (Q1, Q3) (%) | 5.0 (1.2, 12.8) | 10.6 (1.8, 32.2) | −3.468 | <0.001 |
| Whole Lung Lesion Proportions Before Treatment, Median (Q1, Q3) (%) | 5.5 (1.8, 23.9) | 30.3 (16.9, 67.3) | −7.796 | <0.001 |
| Whole Lung Lesion Absorption Ratio, Median (Q1, Q3) | 0.88 (0.56, 0.95) | 0.44 (0.26, 0.76) | 6.803 | <0.001 |

Notes.
[a]For normally distributed variables, mean ± SD denotes statistical measures, with *t*-tests for group differences. Non-normal data are summarized by median (IQR), with rank-sum tests for comparisons.

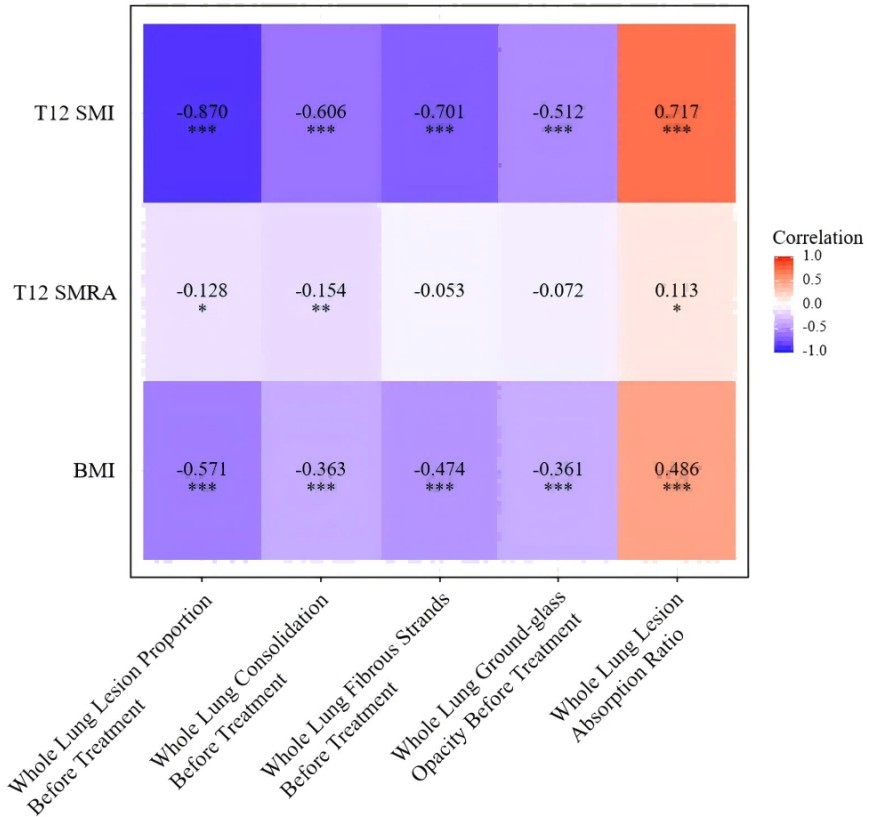

**Figure 4** Univariate correlation analysis of T12 SMI, T12 SMRA, and BMI with pulmonary tuberculosis imaging features. Note: ***, $p < 0.001$; **, $p < 0.01$; *: $p < 0.05$.

**Table 2** Multivariable linear regression analysis of the proportion of whole lung lesions before treatment.

| Variable | $\beta$(95% CI) | p |
|---|---|---|
| T12 Skeletal Muscle Index (T12 SMI) | −4.56 [−5.45, −3.67] | <0.001 |
| T12 Skeletal Muscle Radiation Attenuation (T12 SMRA) | 0.43 [−0.25, 1.12] | 0.215 |
| Age | 0.18 [−0.12, 0.47] | 0.248 |
| Gender | | |
|    Male | reference | |
|    Female | 13.21 [1.40, 25.02] | 0.029 |

**Notes.**
CI, confidence interval; Model included age, gender, T12 vertebral skeletal muscle radiation attenuation, and T12 skeletal muscle index.

Subgroup analyses by gender, age group (≤45 *vs.* >45 years), smoking status, and drinking status consistently demonstrated that lower T12 SMI levels were associated with poor prognosis across all subgroups (Fig. 5). Notably, there were no significant interactions by age, smoking, or drinking status, indicating that the association between low T12 SMI

**Table 3  Multivariable linear regression analysis of the whole lung lesion absorption ratio.**

| Variable | $\beta$(95% CI) | p |
|---|---|---|
| T12 Skeletal Muscle Index (T12 SMI) | 0.036 [0.031, 0.041] | <0.001 |
| T12 Skeletal Muscle Radiation Attenuation (T12 SMRA) | 0.002 [−0.002, 0.006] | 0.335 |
| Age | 0.002 [0.000, 0.004] | 0.014 |
| Gender | | |
|     Male | reference | |
|     Female | −0.017 [−0.086, 0.051] | 0.620 |

Notes.
CI, confidence interval; Model included age, gender, T12 vertebral skeletal muscle radiation attenuation, and T12 skeletal muscle index.

**Table 4  Multivariable logistic regression analysis of factors associated with poor prognosis.**

| Variable | $\beta$ | SE | z | Adjusted OR (95% CI) | p |
|---|---|---|---|---|---|
| Muscle Mass Group | | | | | |
|     Normal Muscle Mass | 0.000 | | | 1.00 (reference) | |
|     Low Muscle Mass | 3.000 | 0.446 | 6.725 | 20.10 [8.81, 51.55] | <0.001 |
| T12 Skeletal Muscle Radiation Attenuation (T12 SMRA) | −0.009 | 0.029 | −0.294 | 0.99 [0.93, 1.05] | 0.769 |
| Age | 0.027 | 0.013 | 2.108 | 1.03 [1.00, 1.06] | 0.035 |
| Gender | | | | | |
|     Male | 0.000 | | | 1.00 (reference) | |
|     Female | 1.126 | 0.526 | 2.140 | 3.08 [1.12, 8.97] | 0.032 |

Notes.
Definition of low muscle mass: T12 SMI < 28.8 cm$^2$/m$^2$ for males and < 20.8 cm$^2$/m$^2$ for females; SE, standard error; OR, odds ratio; CI, confidence interval.

and poor prognosis was robust and consistent regardless of these demographic and lifestyle factors.

## Predictive value of T12 SMI, T12 SMRA, and BMI for poor prognosis in tuberculosis patients

Using the presence of poor prognosis as the outcome variable and pre-treatment T12 SMI, T12 SMRA, and BMI as test variables, an ROC curve was plotted. The analysis revealed that the area under the ROC curve for predicting prognosis was 0.761 (0.690, 0.832) for T12 SMI, 0.602 (0.512, 0.692) for T12 SMRA, and 0.700 (0.619, 0.781) for BMI. T12 SMI demonstrated higher predictive value than T12 SMRA and BMI (See Fig. 6 and Table 5). However, the results should be interpreted with caution due to overlapping confidence intervals and the relatively small sample size (Table S4).

## DISCUSSION

This study investigates the effectiveness of muscle mass-related indicators in predicting the severity and prognosis of active pulmonary TB, with a focus on the T12 SMI as compared to the commonly used BMI. Intergroup analysis revealed that patients with low T12 SMI exhibited more extensive pulmonary lesions, lower post-treatment lesion absorption, and

| Subgroups | Event/No. | | OR (95%CI) | p.value | P for interaction |
|---|---|---|---|---|---|
| Age | | | | | 0.566 |
| ≤45 | 14/159 | | 0.819 (0.710, 0.945) | 0.006 | |
| >45 | 33/150 | | 0.774 (0.695, 0.863) | <0.001 | |
| Gender | | | | | 0.071 |
| Male | 32/216 | | 0.779 (0.697, 0.869) | <0.001 | |
| Female | 15/93 | | 0.611 (0.470, 0.794) | <0.001 | |
| Smoking | | | | | 0.323 |
| No | 33/235 | | 0.798 (0.721, 0.884) | <0.001 | |
| Yes | 14/74 | | 0.697 (0.565, 0.861) | <0.001 | |
| Drinking | | | | | 0.442 |
| No | 40/254 | | 0.760 (0.683, 0.845) | <0.001 | |
| Yes | 7/55 | | 0.611 (0.411, 0.909) | 0.015 | |

0   0.5   1   1.5   2

**Figure 5   Subgroup analyses of the associations between T12 SMI and poor TB prognosis by gender, age group, smoking and drinking status.** Note: T12 SMI was treated as continuous variable in these models. OR, odds ratio; CI, confidence interval.

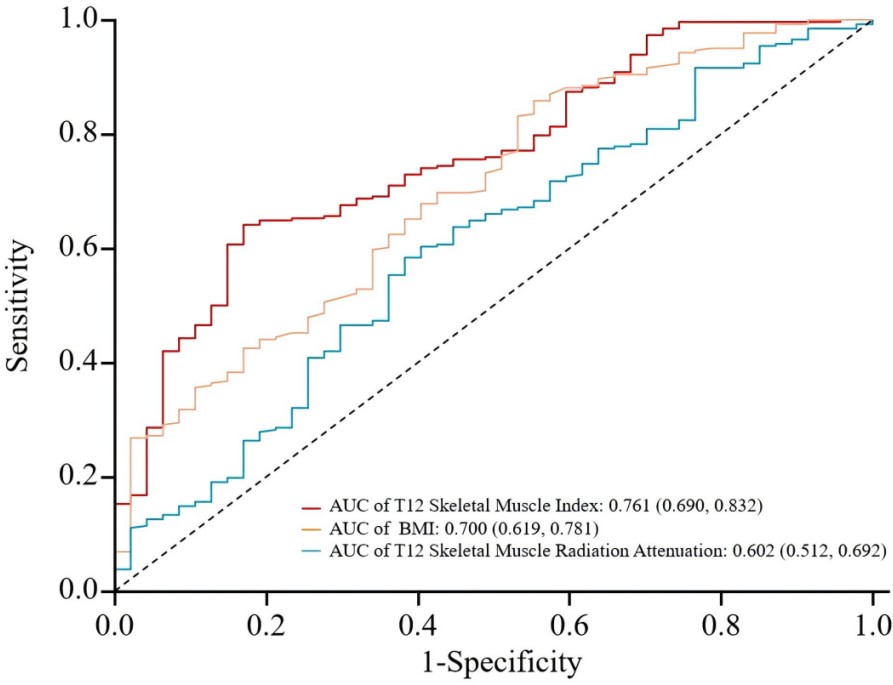

**Figure 6   ROC curves for predicting poor prognosis in tuberculosis patients using T12 SMI, T12 SMRA, and BMI.** ROC, receiver operating characteristic; AUC, the area under ROC curve.

**Table 5  Predictive value of T12 SMI, T12 SMRA, and BMI for poor prognosis in tuberculosis patients.**

| Indicator | Cut-off value[a] | AUC | 95% CI | *p*-value | Sensitivity | Specificity | Maximum youden index |
|---|---|---|---|---|---|---|---|
| T12 Skeletal Muscle Index | 28.351 | 0.761 | 0.690–0.832 | <0.001 | 0.641 | 0.830 | 0.471 |
| BMI | 17.765 | 0.700 | 0.619–0.781 | <0.001 | 0.858 | 0.447 | 0.305 |
| T12 Skeletal muscle radiation attenuation | 42.920 | 0.602 | 0.512–0.692 | 0.013 | 0.584 | 0.617 | 0.201 |

**Notes.**

[a]The cut-off values in these analyses were determined by maximizing the Youden index, a standard method for identifying optimal thresholds in diagnostic tests. BMI, body mass index, calculated by weight in kilograms divided by the square of height in meters; ROC, receiver operating characteristic; AUC, the area under ROC curve; CI, confidence interval.

a higher likelihood of cavitation. While reductions in both T12 SMI and BMI were linked to more severe TB and decreased lesion absorption, the association between T12 SMI and TB severity was stronger. The study also identified that patients with low muscle mass faced nearly 20 times higher odds of poor prognosis than those with normal muscle mass, highlighting T12 SMI was associated with adverse outcomes in TB patients. Moreover, T12 SMI demonstrated superior predictive performance for TB prognosis, significantly surpassing BMI. Consequently, T12 SMI emerges as a simple yet effective indicator for predicting the prognosis of active TB patients, making it invaluable for early clinical assessment.

Several studies have indicated that malnourished TB patients tend to present with more extensive lesions, poorer treatment outcomes, and higher relapse rates (*Hoyt et al., 2019*; *Khan et al., 2006*). Even after standard anti-tuberculosis therapy, these patients remain at an elevated risk of mortality (*Bhargava et al., 2013*). One study linked malnutrition to lower Th1 cytokine concentrations and higher Th2 cytokine concentrations, suggesting that this Th1-to-Th2 shift may weaken the immune defense against *Mycobacterium tuberculosis*, potentially explaining the observed complications (*Sinha et al., 2019*). In this study, lower T12 SMI and BMI were similarly associated with larger pulmonary lesions and reduced post-treatment lesion absorption, supporting these findings. Notably, low T12 SMI was more strongly correlated with extensive TB lesions and poorer treatment outcomes than BMI, likely due to its more accurate reflection of overall muscle mass and, consequently, patient health. However, no significant correlation was observed between T12 SMRA and pulmonary imaging findings, possibly due to the small sample size, which may have resulted in false-negative outcomes. This suggests that T12 SMRA may not play a critical role in TB imaging characteristics.

Consolidation and cavitation on pulmonary imaging are key indicators of active TB (*Nachiappan et al., 2017*). Consolidation signifies persistent infection and active TB lesions within lung tissue, while the high oxygen levels in cavities promote rapid bacterial growth, hindering macrophage function in the necrotic granulomas (*Urbanowski et al., 2020*). In this study, analysis of specific TB imaging features revealed correlations between consolidation and both T12 SMI and BMI. Notably, T12 SMI showed a strong negative correlation with consolidation, and the likelihood of cavitation was higher among patients

with low muscle mass. These findings suggest that T12 SMI may partially reflect TB activity, potentially aiding clinicians in assessing disease severity.

Although low muscle mass is included as a phenotypic criterion for malnutrition by the Global Leadership Initiative on Malnutrition (GLIM) consensus in 2018 (*Cederholm et al., 2019*), specific standards for assessing muscle mass have not been established, leading to variability and inconsistency in clinical practice. *Shin et al. (2020)* found that pectoral muscle mass was a better predictor than BMI for distinguishing between multidrug-resistant tuberculosis (MDR-TB) and drug-sensitive tuberculosis (DS-TB). Similarly, *Shimoda et al. (2023)* demonstrated that the cross-sectional area ($ESM_{CSA}$) and thickness ($ESM_T$) of the erector spinae muscle at the T12 level were significant predictors of mortality in elderly TB patients. However, a subsequent study (*Tanaka et al., 2021*) showed no correlation between $ESM_{CSA}$ and in-hospital mortality in TB patients, casting doubt on the reliability of this metric. Our study takes an innovative approach by utilizing the T12 Skeletal Muscle Index to assess muscle mass and its effect on TB prognosis. The findings confirm that low muscle mass is an independent risk factor for poor prognosis, underscoring the importance of muscle mass in predicting TB outcomes and guiding clinical intervention. Nonetheless, additional research is essential to refine the most effective muscle mass assessment methods for better integration with TB management.

Through ROC curve analysis, we observed that the AUC for T12 SMI was 0.761, which is higher than that of BMI and T12 SMRA. While the difference in AUC values does not reach statistical significance, the ROC curve analysis suggests that T12 SMI offers a more favorable balance between sensitivity and specificity for predicting poor prognosis in TB. This indicates that T12 SMI could be a valuable tool for prognostic assessment, although further studies with larger sample sizes may be needed to confirm the statistical significance of these findings.

Accurately predicting the prognosis of TB requires not only selecting the correct indicators but also precise assessment of TB lesions. Quantitative imaging techniques extract measurable features from medical images, offering a clear representation of disease severity and progression (*Kirby & Smith, 2023*). Most studies on muscle mass assessment have focused on the L3 vertebral level (*Zhang et al., 2023*; *Jin et al., 2023*; *Liu et al., 2024*). However, for tuberculosis patients, routine imaging is typically limited to chest CT scans, which often extend only to the T12 level. Assessing muscle mass at the L3 level in such cases would result in significant data gaps, and extending the scan to cover the L3 level would increase radiation exposure, posing potential risks to patients. Studies have demonstrated that when L3-level muscle data are unavailable, T12 data can be used as a reliable alternative (*Derstine et al., 2018*). As demonstrated in prior research, T12 SMI has been shown to strongly correlate with L3 SMI in various conditions, including tuberculosis, making T12 a reliable surrogate for L3 muscle mass assessment in our study (*Tan et al., 2021*; *Hong et al., 2023*; *Brath et al., 2023*). Many studies *Keicho et al., (2012)*; *Nair et al. (2018)*; *Li et al., (2020)* have relied on qualitative, subjective evaluations of TB lesions on chest CT scans. For instance, *Li et al. (2020)* classified TB imaging features and combined them with patient history and laboratory tests to predict disease severity. However, their grading system was based on the subjective judgment of clinicians and lacked quantitative measures, which
often introduces information bias. In contrast, this study utilized quantitative methods to precisely measure changes in TB lesions following treatment, ensuring that the evaluation of severity and therapeutic efficacy is based on accurate data, thus reducing errors from subjective interpretation.

Interestingly, our logistic regression analysis indicated that female gender was associated with a higher risk of poor prognosis (OR = 3.08). This finding contrasts with some existing literature, which often reports higher TB burdens and worse outcomes in males, potentially due to higher rates of smoking, alcohol use, and occupational exposures among men. However, the observed association in our study may reflect specific characteristics of our patient population, such as differences in healthcare-seeking behavior, comorbidities, or socioeconomic factors that disproportionately affect women in this cohort. Further research is needed to explore the underlying reasons for this gender disparity in TB prognosis.

In clinical practice, clinicians can easily measure T12-based muscle parameters using routine chest CT scans, which are commonly performed for diagnostic purposes. Based on our results, we suggested that the calculation of these T12-based muscle parameters could be integrated into the radiologist's workflow during the diagnostic report generation process, enabling real-time inclusion of T12 muscle parameters in patient reports. This would provide clinicians with immediate access to valuable prognostic information, facilitating early identification of tuberculosis patients at risk for poor outcomes, all while maintaining minimal added cost.

The strengths of this study lie in the quantitative evaluation of muscle mass based on high-resolution chest CT, which quantitatively assessed the nutritional status of patients; it also quantitatively assessed TB severity and prognosis using CT, avoiding errors caused by subjective judgment. Furthermore, this study compared the association and prognostic value of BMI and muscle mass indicators with TB-related outcomes.

Our study has several limitations. Firstly, this was a retrospective, single-center study with a limited sample size, which could have introduced selection bias affecting the results. Given the retrospective nature of this study and its single-center design, the patient cohort may not fully represent the broader TB population. The study cohort may differ from the general population in terms of demographic characteristics (*e.g.*, age, gender), comorbidities (*e.g.*, diabetes, HIV status), and disease severity. Selection bias may arise due to these differences, potentially affecting the generalizability of the results. To assess the potential impact of these factors, a sensitivity analysis was conducted, comparing outcomes across different subgroups based on demographics and clinical characteristics. The findings of this analysis support the robustness of our results, although we acknowledge the limitations inherent in our cohort and encourage further studies in diverse settings to validate these findings. Secondly, neither internal nor external validation could be performed due to the limited sample size. To further validate our findings, prospective, multicenter studies are needed to determine whether T12-based muscle measurements should become a standard metric for tuberculosis risk stratification. Such studies would allow for broader generalization of our results and help confirm the applicability of T12 SMI as a reliable prognostic tool across diverse patient populations. Thirdly, residual confounding by unmeasured factors

such as caffeine intake, alcohol use, nutritional status and physical inactivity could not be ruled out. Since we only measured muscle mass at baseline, it is important to acknowledge that body composition, including muscle mass, may change over the six-month treatment period. Future studies could benefit from incorporating follow-up scans to assess changes in muscle mass over time, which would provide a more comprehensive understanding of its role in prognosis during treatment. However, the consistent results of our regression analysis, using various indicators of disease severity and prognosis, suggest that the observed associations are unlikely to be entirely explained by residual confounding. Fourthly, given the retrospective nature of this study, causal inferences cannot be drawn, and the observed associations should be interpreted with caution.

## CONCLUSION

In conclusion, this study suggests that quantitative evaluation of muscle mass using high-resolution chest CT, particularly the T12 SMI (OR: 20.10, 95% CI [8.81–51.55]), may provide valuable prognostic information for tuberculosis patients, potentially offering advantages over BMI in assessing patient outcomes. T12 SMI was found to be strongly correlated with TB imaging features and treatment outcomes, with low muscle mass identified as an independent risk factor for poor prognosis. However, given the retrospective, single-center design of this study, further prospective, multicenter studies are needed to confirm these findings and explore the potential clinical utility of muscle mass assessment in TB management. Integrating nutritional and body composition evaluations, including T12 SMI, into clinical practice could enhance early interventions and improve patient outcomes.

## ACKNOWLEDGEMENTS

The authors would like to thank Dr. Jie Lin of the Yongjiang Laboratory for his professional guidance in statistics.

### Funding

This work was supported by Ningbo Clinical Research Center for Medical Imaging (No. 2021L003) and NINGBO Leading Medical & Health Discipline (No. 2022-S02). The funders had no role in study design, data collection and analysis, decision to publish, or preparation of the manuscript.

### Grant Disclosures

The following grant information was disclosed by the authors:
Ningbo Clinical Research Center for Medical Imaging: 2021L003.
NINGBO Leading Medical & Health Discipline: 2022-S02.

### Competing Interests

The authors declare there are no competing interests.

## Author Contributions

- Ankang Huang conceived and designed the experiments, performed the experiments, analyzed the data, prepared figures and/or tables, and approved the final draft.
- Yuyao Zhang analyzed the data, prepared figures and/or tables, and approved the final draft.
- Qi Dai performed the experiments, authored or reviewed drafts of the article, and approved the final draft.
- Jingfeng Zhang analyzed the data, authored or reviewed drafts of the article, and approved the final draft.
- Jianjun Zheng conceived and designed the experiments, authored or reviewed drafts of the article, and approved the final draft.

## Human Ethics

The following information was supplied relating to ethical approvals (i.e., approving body and any reference numbers):

Ethical Approval was obtained from Medical Ethics Committee of the Ningbo NO.2 Hospital (NBEY-2023-064).

## Data Availability

The raw data is available in the Supplemental Files.

## Supplemental Information

Supplemental information for this article can be found online at http://dx.doi.org/10.7717/peerj.19147#supplemental-information.

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
