# Peer review of "Quantitative evaluation of muscle mass based on chest high-resolution CT and its prognostic value for tuberculosis: a retrospective study"

_PeerJ, doi:10.7717/peerj.19147_

## Round 0.1 · original submission · Major Revisions

1. For the statistical analysis, do you mean multivariable (multiple covariates) or multivariate (multiple responses) logistic regression?

2. If multivariable, you should report adjusted odds ration (aOR), not just OR.

·

Basic reporting

English use is satisfactory.
References relevant and up to date.
Tables incorrectly titled and incorrectly described in the text.
The results presented in the manuscript, after a re-interpretation by the reviewer, seem to be partially acceptable, but there is a lack of clarity on the outcome used and type of model constructed in Tables 2, 3 and 4.

Experimental design

Research questions are defined, relevant and meaningful.
Technical aspects of the data analysis methods are inadequately described.

Validity of the findings

Cannot be properly judged as some of the analytical methods are not adequately described and some of the tables are confusing, leading to considerable confusion regarding their interpretation.

Additional comments

This paper examines the association between muscle mass and characteristics of (pulmonary) tuberculosis patients based on high-resolution CT, and the performance of muscle mass in predicting anti-tuberculosis treatment outcomes. There are a number of issues in this manuscript that should be addressed. These are detailed below, largely in the order in which they appear in the manuscript.
There appear to be two slightly different versions of the Abstract included in the “Manuscript to be reviewed”, one referring to linear regression yielding odds ratios, and the other in which a revision has been made to refer to logistic regression.

The Results section of the Abstract states that “ROC curve analysis further confirmed the superiority of T12 SMI in prognostic prediction, with an area under the ROC curve (AUC) of 0.761 (95% CI: 0.690-0.832), which was significantly higher than the AUC of BMI (0.700, 95% CI: 0.619-0.781).” However, no formal test showing a statistically significant difference between these two ROC curves is presented in the manuscript.

In the Statistical Analysis section of the Methods, it is stated that multivariable linear regression was used to study the association between the skeletal muscle index and imaging indictors. These linear regression models are not shown in the Results section. (See also comments below.) Rather, these comparisons are shown in the bivariate correlations in Figure 4.

Section 1.4 Prognosis Evaluation in the methods section states: “Prognosis was classified into five categories…”. It seems that, strictly speaking, these are classifications of the outcome rather than of prognosis.

In the Results section mention is made of the “proportion of pre-treatment whole-lung fibrotic lesions” and the “proportion of lesion absorption”. These are also mentioned in the Methods section. Table 1, however, refers to “whole lung lesion ratio” and “whole lung lesion absorption ratio”. Are the proportion and ratio variables referring to the same parameters? It seems that the table is, in fact, not referring to proportions as there is a value of 3.03 for whole lung lesion ratio before treatment among the low muscle mass group. The value of 3.03 suggests a ratio rather than a proportion unless the values are presented as a percentage, in which case 3.03 could be a proportion. However, no explanation is given.

On the other hand, what are described as the results of linear regression in Tables 2 and 3 appear to be the results of logistic regression as they show odds ratios. Even so, if they are actually the results of logistic regression, the question arises of what exactly are the outcome variables in these models. Logistic regression that leads to odds ratios requires that the outcome variable is binary, yet the titles of these figures refers to proportions, which are not binary – that is, each patient has his/her own value of these proportions (of whole lung lesion before treatment and of whole lung lesion absorption following treatment, in Tables 2 and 3 respectively).

A similar anomaly is seen in Table 4, whose title describes it as the results of “Multivariate [multivariable?] Cox regression of factors associated with poor prognosis”. While poor prognosis as defined in the manuscript is indeed a binary outcome, there is no indication of why such a time-to-event model should be used, and, furthermore, the results are shown as odds ratios, suggesting that, in fact, they are the results of a binary logistic regression model.

Given that the models are actually all logistic regression models, there is the question of the extent to which the variables T12 SMI and T12 SMRA are themselves correlated. This is not shown in the manuscript. As one would expect a reasonable level of correlation among these indicators of muscle mass, including them both in the same model is likely to give misleading estimates of their levels of association with the outcome. (However, including both variables together does, indeed, indicate that T12 SMI is more closely associated with the outcome than T12 SMRA.)

It is noted that Table 4 includes the variable “T12 Skeletal Muscle”. Does this refer to T12 SMI or T12 SMRA?

In Table 5, the basis for choosing those particular cut-off values should be stated.

·

Basic reporting

see below

Experimental design

see below

Validity of the findings

see below

Additional comments

General Overview: The manuscript addresses an important area of tuberculosis (TB) research: the prognostic value of muscle mass assessment using high-resolution chest CT in patients with TB. The study presents findings that highlight the relationship between skeletal muscle mass and TB outcomes and suggests that the T12 Skeletal Muscle Index (SMI) is a more reliable predictor of prognosis than Body Mass Index (BMI). While the research is relevant and aligns with the field's goals, several weaknesses and methodological gaps need to be addressed to enhance the manuscript’s robustness and scientific rigor.
1. Basic Reporting
Clarity and Language: The manuscript generally uses professional language but should be better phrased to avoid confusion. Some sentences are unnecessarily complex or vague, which could hinder understanding. For example, in the Introduction, the phrase "to provide more evidence-based guidance for clinical practice" is overly general. It would be more precise to state how this study’s findings will specifically guide clinical decisions regarding TB management.
• Suggested Revision: Simplify complex sentences, for example, rephrase "providing more evidence-based guidance for clinical practice" to "informing clinical interventions for TB patients by identifying those at higher risk of poor prognosis based on muscle mass evaluation."
Literature Context and References: The literature review in the Introduction does not adequately cover recent studies that link muscle mass with infectious disease outcomes, nor does it cite enough TB-specific studies comparing different prognostic factors. This limitation could weaken the paper’s context-setting.
• Suggested Revision: Integrate more contemporary studies that explore the relationship between sarcopenia, malnutrition, and TB, particularly those addressing the predictive capabilities of muscle mass vs. BMI in various conditions.
2. Experimental Design
Study Design and Justification: The rationale for using T12 SMI as the muscle mass assessment metric requires better justification. While this index is clinically relevant, other vertebral levels or muscle groups could also be predictive. The manuscript should include a discussion on why T12 was chosen over alternatives and the potential implications.
• Suggested Revision: Explain why T12 was selected as the site for skeletal muscle assessment, referencing relevant studies supporting this choice. Also, discuss potential limitations in comparison to other commonly used metrics like the psoas muscle index or L3 level.
Patient Selection and Bias Concerns: The inclusion and exclusion criteria introduce possible selection bias, given the retrospective single-center nature of the study. The manuscript should address how the patient cohort represents the broader TB population and whether any potential biases may have influenced the findings.
• Suggested Revision: Add a paragraph discussing selection bias and note how the study cohort might differ from the general TB patient population in terms of demographics or comorbidities. Consider including a sensitivity analysis if data permits.
Data Analysis and Statistical Methods: The statistical methods are generally appropriate; however, the manuscript does not show how potential confounders were handled. For example, the adjustment for covariates such as age and sex in multivariable models is mentioned, but the reasoning behind choosing specific covariates is not explained.
• Suggested Revision: Provide an explanation of the choice of covariates included in multivariate analyses. Consider including interaction terms to explore whether the relationship between T12 SMI and prognosis varies by key factors such as age or sex.
3. Validity of the Findings
Interpretation of the Results: The manuscript often overstates the findings, particularly when claiming the superiority of T12 SMI over BMI. While the ROC analysis showed a higher AUC for T12 SMI, the actual difference is modest. This overstatement could mislead readers regarding the clinical significance of the findings.
• Suggested Revision: Revise the language to reflect the modest differences in predictive ability. Emphasize that while T12 SMI may offer advantages over BMI, the results should be interpreted with caution due to overlapping confidence intervals and the relatively small sample size.
Correlation and Causation: There is a tendency to imply causation where only association has been demonstrated. The study design does not allow for causal inferences about muscle mass and TB outcomes.
• Suggested Revision: Reframe statements that imply causality. For example, "Low T12 SMI is a risk factor for poor prognosis" should be changed to "Low T12 SMI is associated with poor prognosis." Emphasize that the study's retrospective nature limits causal interpretation.
Potential Confounding Variables: The exclusion criteria omitted patients with conditions that could significantly affect chest CT results, such as comorbidities. However, this approach does not account for other factors like physical activity levels or diet, which could influence muscle mass and TB outcomes.
• Suggested Revision: Acknowledge the limitations due to potential residual confounding factors that were not measured, such as physical inactivity or nutritional status, which may have affected the study's findings. Including this discussion will provide a balanced view of the study's results.
4. Specific Recommendations for Each Section
Introduction
• Expand the discussion on muscle mass as a prognostic factor, drawing on more diverse studies in both infectious and non-infectious diseases.
• Clarify the study’s objective to emphasize its unique contribution to current knowledge, particularly in comparison with previous studies focusing solely on BMI.
Methods
• Justify the choice of T12 as the vertebral level for muscle mass evaluation.
• Elaborate on how the AI-assisted quantification of lung lesions was validated and whether there was any inter-observer variability.
Results
• Provide a more nuanced interpretation of statistical findings, emphasizing the clinical implications of the observed differences in AUC values between T12 SMI and BMI.
• Add subgroup analyses if possible, to show whether the associations between muscle mass and TB prognosis differ across age groups or other subcategories.
Discussion
• Acknowledge alternative explanations for the observed associations, such as the possibility that low muscle mass may be a marker of overall frailty or comorbidities rather than a direct cause of poor TB outcomes.
• Address the limitations of the study’s retrospective nature and single-center design more comprehensively.
Conclusion:
• Avoid definitive statements suggesting T12 SMI should replace BMI in clinical practice based solely on this study’s findings. Instead, call for prospective, multicenter studies to confirm these results.

---

## Round 0.2 · Major Revisions

The manuscript would benefit from input by a (bio)statistician.

·

Basic reporting

Some mistakes, as indicated in my report below.

Experimental design

Design is satisfactory.

Validity of the findings

Some issues are mentioned in my report, below.

Additional comments

This revised manuscript is much improved compared with the previous version. However, there are still some issues that should be dealt with. These will, as before, be listed largely in the order in which they appear in the manuscript.

Abstract and Tables 1, 2 and 3:
Lines 33-36: “Multivariable linear regression analysis revealed that after adjusting for age, gender, and T12 skeletal muscle radiation attenuationÿT12 SMRAÿ, T12 SMI remained significantly correlated with the percentage of lung lesion absorption (OR: 1.037, 95% CI: 1.031, 1.042) and lesion proportion (OR: 0.634, 95% CI: 0.580, 0.693).”

Linear regression analysis does not yield odds ratios. It seems that the beta coefficients in Tables 2 and 3 of -0.456 and 0.036, have been exponentiated to give these values of 0.643 and 1.037. If, indeed, the models are linear regression models, then the beta coefficients stand as they are and indicate the change in proportion of whole lung lesions before treatment and change in whole lung absorption ratio, respectively, associated with a unit increase in the value of T12 SMI etc.

However, the units of while lung lesion proportion have been changed in the revised manuscript. They are no longer 0.55 and 3.03 in the normal muscle mass group and low muscle mass group, respectively, as given in the original manuscript using apparently “1/10 units”, but are now 5.5% and 30.3%. The coefficients in Table 2 appear to pertain to the original 1/10 units and now will not be correct. If my interpretation is correct, these coefficients should be increased by a factor of 10 giving, for example, -4.56 for the coefficient of T12 SMI. Similar changes would be required for the other coefficients in this table.

Methods:
Lines 155-156: “Prognosis outcome was classified into five categories: cure, treatment
completion, treatment success, treatment failure, and death.”
My understanding is that these are the actual outcomes, not prognosed outcomes.

Results:
Lines 237-239: “Additionally, age (OR = 1.028, 95% CI: 1.002, 1.055, p < 0.035) and gender (OR = 3.084, 95% CI: 1.116, 8.968, p < 0.032) were also identified as independent risk factors associated with the prognosis in tuberculosis patients. See Table 4.”

Perhaps this should be revised slightly to “Additionally, age (OR = 1.028, 95% CI: 1.002, 1.055, p < 0.035) and female gender (OR = 3.084, 95% CI: 1.116, 8.968, p < 0.032) were also identified as independent risk factors associated with the prognosis in tuberculosis patients (Table 4).”

Lines 240-241: “Subgroup analyses by gender, age group, smoking and drinking status showed that in all subgroups, lower T12 SMI levels were consistently associated with higher TB risk (Figure 5).”

I think “TB risk” is not appropriate. All patients had active TB. Perhaps the sentence could be revised to “Subgroup analyses by gender, age group, smoking and drinking status showed that in all subgroups, lower T12 SMI levels were consistently associated with poor outcome (Figure 5).”

Table 3: It is sometimes a little questionable whether modelling a ratio using linear regression is quite appropriate as ratios are not normally distributed. However, as all ratios in this study are less than unity and not widely divergent, this issue is not of much importance here, and linear regression can be used.

Discussion:
Line 304-307: “The results showed that the AUC for T12 SMI was 0.761, significantly higher than that of BMI and T12 SMRA, indicating that T12 SMI has a high predictive performance in TB prognosis and can be used as an independent indicator for accurately predicting patient outcomes.”

The data do not show that the AUC for T12 SMI is significantly higher that that for BMI. Nevertheless, even though the difference may not be statistically significant (Is the Table S5 as shown in the responses to reviewers going to be included among the supplementary material?) meaning that we are a little less confident about their being a real difference in AUC, the ROC curve clearly shows that the sensitivity-specificity combination(s) provided by T12 SMI would be much more appropriate for making useful prediction of poor prognosis than would those provided by BMI. I think this point should be included in the discussion.

Figure 6: The labels of the y and x variables appear to be incorrect. The y-axis shows the sensitivity and the x axis shows 1-specificity.

Table 5. The headings of the right-most three columns should indicate that the Youden index shown is the maximum Youden index and that the sensitivity and specificity are those obtained when the cut-point of the indicator corresponds to that indicated by the maximum Youden index.

---

## Round 0.3 · Minor Revisions

The reviewer identifies an inconsistency between the units of whole lung lesion proportions reported in Table 1 and the coefficients in Table 2. To maintain consistency across the manuscript, the authors should clarify the units (percent, tenths, or decimal fractions) and adjust the Table 2 coefficients if necessary.

·

Basic reporting

Satisfactory, but see comments below

Experimental design

Satisfactory

Validity of the findings

One issue of concern. Please see below.

Additional comments

This second revision of the manuscript has adequately addressed all but one of my comments on the previous version,
The remaining issue is the value of the coefficients in Table 2. I accept that the coefficients may be appropriate to the measurement of whole lung lesion proportions in the units of “tenths”. But they are surely not appropriate to the measurement in percent, as shown in Table 1.
We learn from Table 1 that the median proportions of whole lung lesion before treatment in the normal and low muscle mass groups are approximately 5.5 and 30.3 respectively. Also, the mean values of T12 SMI in the two groups are respectively 31 and 25. Assuming these are representative of the two of the points in a scatter plot of whole lung proportion against T12 SMI then the slope between them would be something like (5.5 - 30.3)/(31-25) or approximately -25/6 = -4.2. Only if the units of whole lung proportion were in “tenths” would the slope be approximately -2.5/6 = -0.42 (i.e., close to the coefficient of -0.456 as shown in Table 2). Note that the coefficients in a linear regression model have units.
So, presumably, the units of whole lung proportion in Table 1 should be changed to units of “tenths” (i.e., 0.55 and 0.303), or the coefficients in Table 2 increased by a factor of 10, to maintain consistency through the manuscript.
Lines 115-6 might be written more explicitly something like “The whole lung lesion proportion before treatment was defined as the percent [or “tenths” or “decimal fraction”] of the total lung volume occupied by the lesion.” That is, avoid the term “ratio” but be clear as to whether the proportion is expressed as a percentage or in units of “tenths” or, indeed, as a decimal fraction (which would require the coefficients in Table 2 to be divided by 10!).
If the authors think I am mistaken, I should welcome an explanation of where they feel I have misunderstood the data.

---

## Round 0.4 · Minor Revisions

Both reviewers commended your study and agreed that the manuscript has substantially improved clarity. Reviewer 2 suggests the following for further revision:
o Further detailing the AI exclusion specifics,
o Summarizing subgroup analysis results in the text,
o Providing interpretive context for unusual findings (e.g., the female OR),
o Fine-tuning rationale for T12-based muscle measurement and the single-centre limitations.

·

Basic reporting

No comment

Experimental design

No comment

Validity of the findings

No comment

Additional comments

The authors have adequately addressed my comments concerning the previous version.

·

Basic reporting

Introduction
1. Rationale for T12 (Succinctly Introduce)
o In the final paragraph that states your research objectives, you might add a short phrase explaining why T12 SMI is a logical surrogate (since L3 is often not captured on chest CT). This helps tie the background about muscle mass to your specific methodology right away.
2. Population & Comorbidity Focus
o You mention that existing muscle mass–TB research often focuses on elderly patients or those with comorbidities. Since you excluded diabetes, HIV, COPD, etc., it may help to emphasize that your study intentionally focused on drug-sensitive TB patients without major comorbidities—thus clarifying the population scope for your results.

Experimental design

Methods
Study Population
• Explicitly Describe the 24 AI-Error Exclusions
o While you reference 24 excluded cases due to quantification errors in the main text, ensure the exact total excluded for each reason is clearly stated in a single place. This helps avoid confusion on the final N=309 and how each category of exclusion was applied.
Imaging Analysis
• Blinding of Radiologists
o You note that two radiologists verified AI outputs. Clarify if they were blinded to each other’s assessments or to patient data. If they were fully blinded to patient outcome, stating so explicitly adds rigor to your methods.
Statistical Analysis
• Logistic Regression Model Details
o You mention adjusting for age, sex, T12 SMRA, etc. Consider stating how many poor outcomes there were (death/treatment failure) to confirm an adequate event-per-variable ratio. That can be a simple statement like: “Among 309 patients, X experienced poor outcomes.”

Validity of the findings

Results
1. Subgroup Analysis
o You present a forest plot (Figure 5) for subgroup analyses (gender, age groups, smoking, drinking). A short text summary—highlighting any notable differences or if they all aligned similarly—would improve clarity. For instance, “Lower T12 SMI was consistently associated with poor prognosis in all subgroups, with no significant interaction by age or smoking status.”
2. Interpretation of the Female Gender OR
o The logistic regression suggests that female gender is associated with a higher risk of poor prognosis (OR≈3.08). A brief interpretative statement—whether this aligns with existing TB literature or not—would be helpful, as many studies show men may have higher TB burdens. If you suspect your hospital’s patient profile or certain sample characteristics, a short explanation can help contextualize.
Discussion
1. T12 vs. L3 Justification
o You have a clear mention that T12 can be used as a surrogate for L3 muscle mass. It might be worth briefly highlighting any references or prior studies that specifically validated T12 in other conditions or in TB (if available). You do mention some references, but adding one more sentence explaining “T12 SMI strongly correlates with L3 SMI in prior research” (if that is indeed the case) would reinforce your rationale.
2. Future Implementation
o Because you show that T12 SMI outperforms BMI for prognosis, mention how, in practice, clinicians can quickly measure T12-based muscle parameters from a routine chest CT with minimal added cost. This could emphasize the real-world feasibility of your approach.
3. Limitations
o You do a good job of acknowledging potential bias from a single-center, retrospective design. Also add a note on whether “body composition changes over the six-month treatment,” but you only measured muscle mass at baseline. Not a requirement to re-scan T12, but acknowledging that muscle mass might evolve during therapy is helpful context.
Conclusion
1. Explicitly State the OR
o Your conclusion mentions that T12 SMI is an independent risk factor for poor prognosis. You might restate the approximate OR (around 20) to highlight the magnitude of that effect.
2. Future Prospects
o Emphasize that prospective, multicenter validation could determine if T12-based muscle measurement should become a standard metric for TB risk stratification. That final note would show how your findings could be generalized in larger cohorts.

---

## Round 0.5 · accepted · Accept

All reviewers' comments have been adequately addressed

·

Basic reporting

No further comments

Experimental design

No further comments

Validity of the findings

No further comments

Additional comments

All previous comments have been adequately addressed.